

# Decline in coral cover and flattening of the reefs around Mauritius (1998–2010)

Jennifer A. Elliott[1], Mark R. Patterson[1], Caroline G. Staub[2], Meera Koonjul[3] and Stephen M. Elliott[4]

[1] Marine Science Center, Northeastern University, Nahant, MA, United States of America
[2] Institute of Food and Agricultural Sciences International Programs, University of Florida, Gainesville, FL, United States of America
[3] Albion Fisheries Research Center, Ministry of Fisheries, Petite Rivière, Mauritius
[4] Woods Hole Oceanographic Institution, Woods Hole, MA, United States of America

## ABSTRACT

Coral reefs are degrading through the impacts of multiple anthropogenic stressors. How are coral reef communities going to change and how to protect them for future generations are important conservation questions. Using coral reef data from Mauritius, we examined changes in cover in 23 benthic groups for a 13-yr period and at 15 sites. Moreover, we determined which land-based stressor out of four (human population, agriculture, tourism, rainfall) correlated the most with the observed changes in coral reef cover. Among the stony corals, *Acropora* corals appeared to be the most impacted, decreasing in cover at many sites. However, the non-*Acropora* encrusting group increased in cover at several sites. The increase in abundance of dead corals and rubble at some sites also supported the observations of stony coral decline during the study period. Additionally, the decline in stony corals appeared to be more pronounced in second half of the study period for all sites suggesting that a global factor rather than a local factor was responsible for this decline. There was little change in cover for the other benthic groups, some of which were quite rare. Human population was significantly correlated with changes in coral reef cover for 11 sites, followed by tourism and agriculture. Rainfall, a proxy for runoff, did not appear to affect coral reef cover. Overall, our results showed that there has been a decline of stony coral cover especially the ones with complex morphologies, which in turn suggest that coral reefs around Mauritius have experienced a decline in habitat complexity during the study period. Our study also suggests that humans are an important factor contributing to the demise of coral reefs around the island.

Corresponding author
Jennifer A. Elliott,
j.ahking.elliott@gmail.com

## INTRODUCTION

Coral reefs worldwide are declining through the impacts of many different types of natural and anthropogenic stressors (*Pandolfi et al., 2003*). The planet's climate is changing at a pace faster than ever observed before primarily due to the extensive burning of fossil fuels. As a result, the oceans are warming (*IPCC, 2014*) and stony corals, the foundation species of coral reefs, have been observed to bleach and die on a large scale (*Roberts et al., 2002*; *Gardner et al., 2003*; *Pandolfi et al., 2003*; *Bellwood et al., 2004*; *Graham et al., 2006*;

*Bruno & Selig, 2007*; *Heron et al., 2016*; *Hughes et al., 2017*). Severe bleaching events are happening five times more often today than in the 1980s (*Hughes et al., 2018*).

Over 60% of the world's coral reefs have been estimated to be threatened by local stressors arising from anthropogenic activities (*Burke et al., 2012*). Land clearing for agriculture and coastal urbanization have led to sediments, nutrients, and pollutants being discharged through runoff into these coastal ecosystems (*Smith et al., 1981*; *Fabricius, 2005*; *Markey et al., 2007*; *Brodie et al., 2012*). Overfishing has also caused significant changes in ecological communities and contributed to the collapses of some coastal ecosystems (*Jackson et al., 2001*).

How are coral reef communities going to change in the face of multiple anthropogenic stressors and what makes some species more resilient than others are critical conservation questions. Inherent biological traits (e.g., morphology, growth rate and reproductive mode) and physiological mechanisms of coral species are thought to play an important role in their capabilities to respond to repeated assaults from multiple stressors (*Edmunds & Gates, 2008*; *Darling et al., 2012*). For example, a study on coral community structure reported that coral colonies with branching morphologies bleached and died before those with massive and encrusting morphologies after a bleaching event (*Loya et al., 2001*). Theoretical work has also shown that at the same flow rate, an organism with a flat morphology will have a higher mass transfer or diffusion rate (of gases and nutrients to and from the organism) than an organism with a branched morphology (*Patterson, 1992*). This could in turn also lead to faster removal of harmful superoxides and toxic radicals produced during stressful thermal events (*Lesser, 1997*), providing a competitive advantage to species with a flat morphology.

Physical external factors can also play an important role in helping coral reef organisms face multiple stressors. Localized upwelling (*Glynn & D'Croz, 1990*), water-flow rates and flushing of toxins have been reported to help stony corals weather thermal stresses (*Nakamura & Van Woesik, 2001*; *Nakamura & Van Woesik, 2005*). Decreased light stress, e.g., in the form of shading, cloud cover or turbidity can also reduce some of the impacts from stressors (*Hoegh-Guldberg, 1999*; *Moothien Pillay, Terashima & Kawasaki, 2002*; *Jokiel & Brown, 2004*; *Coelho et al., 2017*).

Both global and local efforts are needed to conserve coral reefs worldwide as they continue to degrade because of increasing human population and their activities (*Mora, 2008*; *Sandin et al., 2008*; *Smith et al., 2016*; *Crane et al., 2017*). Long-term coral reef monitoring and studies disentangling the impacts of diverse anthropogenic activities are especially important to inform conservation actions. We investigated the effects of four land-based stressors on the coral reefs around Mauritius. This Indian Ocean island has one of the densest human population on earth, with 618 individuals $km^{-2}$ in 2017 (http://data.worldbank.org). Moreover, a large area of the island has been converted to agricultural land and the island is a prime tourist destination in the Indian Ocean. The two primary goals of this study were to examine changes in coral reef cover over a 13-yr period, and to investigate the relationship between the land-based stressors and changes in benthic communities around Mauritius.

## MATERIALS AND METHODS

### Study sites

This study was carried out at the southwestern Indian Ocean island of Mauritius (20°10′S, 57°31′E; Fig. 1A). The island has a surface area of 1,869 km$^2$ and possesses 240 km$^2$ of reef habitats (*Turner & Klaus, 2005*). It also lies within the cyclonic belt and is visited by cyclones seasonally, typically from January to March. The land area is also divided into 42 watersheds (or catchment areas; Fig. 1B) (*Proag, 1995*). A total of 15 sites, 11 back- and four fore-reef sites, were investigated. Back-reef sites (circles) were found all around the island and fore-reef sites (stars) only on the west coast, which is sheltered from the prevailing southeast trade winds (Fig. 1B). The reefs around Mauritius are broken up by natural breaks that result in a series of distinct lagoons. Back-reef sites were found in the lagoons that are <3 m deep and 1–8 km wide. Additionally, the island has one 10 km long barrier reef on the southeast coast where one of the back reef site was found. This barrier reef is also next to a deep channel (15–30 m) that flushes seawater offshore (Fig. 1B) (*Turner & Klaus, 2005*). The fore-reef sites were found outside the lagoons in 8–20 m deep waters.

Some 160 stony corals species have been reported in Mauritius (*Moothien Pillay et al., 2002*). The back-reefs of Mauritius consist mainly of branching and tabular *Acropora*, massive *Porites*, foliaceous *Montipora* and *Pavona*, and sand consolidated with seagrass. The narrow reef flats are made up mostly of dead coral platform, stony corals and macroalgae, and the outer slopes have a spur and groove system that is colonized by a diverse range of hard and soft coral genera (e.g., *Pocillopora*, *Favia*, *Porites*, *Sarcophyton* and *Lobophyton*). For a more comprehensive description of the coral reefs of Mauritius, see *Turner & Klaus (2005)*.

### Datasets

#### Benthic communities

Benthic coral reef monitoring data spanning over 13 years (1998 to 2010) were obtained from the Albion Fisheries Research Centre (AFRC), Ministry of Fisheries, Government of Mauritius. The data were collected from three 20 m long transects at each site; each transect was marked by two metal stakes. The line intercept method (*Kaiser, 1983*) was used to survey these transects. A measuring tape was laid down along each transect; all benthic groups touching the tape were recorded and the number of centimeters that they covered along the tape was also recorded. The number in centimeters for each benthic group was then converted to percentages such that each transect would be equal to 100%. For example, if sand was found from 0 cm to 23 cm, then sand would be equal to $(23 - 0)/(20 * 100) * 100 = 1.15\%$ of the transect. The three transects were considered as three replicates for each site.

The data were originally collected twice a year, once in summer (December to May) and once in winter (June to November). However, later in the study period, data were sometimes only collected once a year and not all sites were surveyed each year during the study period (Table S1—study years by site). During this 13-yr period, the data were collected by a small group of staff from the AFRC, which minimized observer error in the data.

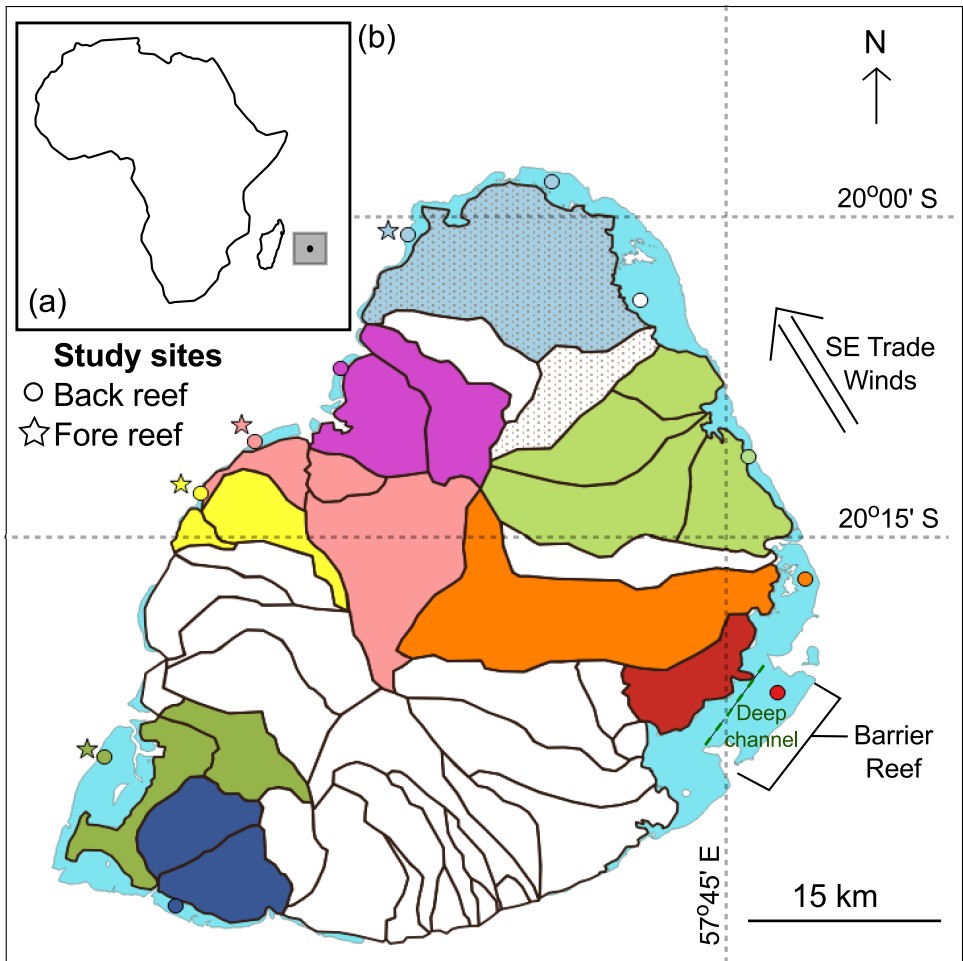

**Figure 1  Mauritius.** (A) The island of Mauritius in the southwestern Indian Ocean. (B) Map of the island showing back- (circles) and fore- (stars) reef sites. Land area is divided into several watersheds (catchment areas). Study sites were connected to adjacent watersheds(s) and are color-coded identically. The back-reef site colored white is connected to adjacent shaded watersheds.

The back-reef sites surveyed were: Albion (ALB), Anse La Raie (ALR), Baie du Tombeau (BDT), Bambous Virieux (BVX), Bel Ombre (BOM), Belle Mare (BME), Île aux Bénitiers (BEN), Poudre d'Or (PDO), Pointe aux Sables (PTE), Trou aux Biches (TBI), Trou d'Eau Douce (TDO). The four fore-reef sites were: Albion (ALB), Île aux Bénitiers (BEN), Pte aux Sables (PTE) and Trou aux Biches (TBI). The back-reefs sites were in waters <3 m deep; the fore-reef sites ALB, BEN, PTE and TBI were set up respectively at 10 m, 10 m, 8 m and 12 m deep.

The benthic cover data were recorded as one of the following 23 groups: *Acropora* branching, *Acropora* digitate, *Acropora* encrusting, *Acropora* submassive, *Acropora* tabulate, non-*Acropora* branching, non-*Acropora* encrusting, non-*Acropora* foliaceous, non-*Acropora* submassive, non-*Acropora* massive, fungiid coral, turf algae, macroalgae, coralline algae, *Millepora* spp., sponges, soft corals, zoanthids, other live, dead corals, rubble, sand and

rock. Dead corals consisted of calcium carbonate coral skeletons that were more or less intact; they would eventually become rubble when they finally break down into smaller pieces.

### Land-based stressors

Four land-based stressors on the coral reefs were investigated: human population, tourism, agriculture, and rainfall. Spatial data for the selected stressors were gathered for the period of 1998 to 2010 so that they would match the time period for which the benthic data were available. Data for the first three stressors were obtained from the governmental organization 'Statistics Mauritius' (http://statsmauritius.govmu.org). Rainfall data were obtained from the monthly meteorological summaries (Mauritius Meteorological Services) that were made available by the Mauritius Sugar Industry Research Institute (MSIRI) archives department.

Data for human population, agriculture and tourism were spatially reorganized using a geographic information system (GIS; QGIS Lyon 2.12) such that each stressor would have one value per watershed per study year (Fig. 1B). The human population data were organized as the total number of people per watershed per year. The agricultural data were organized as total surface area under cultivation per watershed per year. The agricultural data consisted of both land under sugarcane and other food crop cultivation during the study period. For the tourism data, a tourism index was created because the available data were rather coarse; only the number of hotels per watershed per year and total number of tourist nights for the whole island per year were available. This tourism index was created by dividing the number of hotels per watershed by the total number of hotels for the whole island and multiplied by total number of tourist nights. The resulting index gave an approximate number of tourist nights per watershed per year.

A more extensive spatial analysis was done on the rainfall data using ArcGIS 10 (*ESRI, 2011*). Simulated monthly rainfall totals generated by *Staub, Stevens & Waylen (2014)* were averaged across months and then by watershed for each study year. In their study, monthly rainfall totals (2000–2011) from 85 stations, a digital elevation model (DEM) and location-specific landscape characteristics were used to calculate linear relationships between mean annual rainfall and landscape characteristics on Mauritius. The time and space-varying nature of the modeled relationships were then explored by estimating separate models for each month (*Staub, Stevens & Waylen, 2014*). Finally, monthly rainfall totals for 1998 to 2011 were generated by removing the variability characteristic to each month from the observed data, interpolating these values across space using ordinary kriging, and adding these back to the spatial mean model for the corresponding month (*Staub, Stevens & Waylen, 2014*). Since rainfall was a proxy for runoff in this study the rainfall data (mm) per watershed per year was multiplied by their respective watershed areas ($m^2$) and divided by 1,000 to obtain a volume of rainfall per watershed ($m^3$) per year.

### Linking the stressor effects arising within watersheds to study sites

Landscape characteristics including mountains and riverbeds were used to determine which watershed(s) drained into which lagoons. This was done as multiple back-reef sites were found in different lagoons and were thus impacted by different levels of land-based
stressors. Study sites and their respective watersheds were color-coded and are displayed in Fig. 1B. When sites from different lagoons shared a common watershed, one of the sites was colored white and the watersheds to which it was linked were shaded (Fig. 1B). The fore-reef sites were linked to the same watersheds as their complementary back-reef sites. Table S2 provides the long-term mean of each of the four local stressors by site for the period spanning 1998 to 2010.

## Statistical analysis

All statistical tests in this study were done using R (*R Core Team, 2015*).

### Trends in benthic cover for 1998–2010

Temporal changes in benthic cover on the back- and fore-reef were examined. A stacked bar graph was made for 1999 to show how the benthic cover varies by site. The benthic groups were aggregated into eight groups as it was difficult to visualize 23 groups (23 colors). The eight groups were: *Acropora* corals, non-*Acropora* corals, macroalgae, turf algae, coralline algae, dead corals, other living organisms and non-living organisms (included rubbles).

A mean percent cover per year for each benthic group was calculated for each season and for each study site. This was done by averaging data from the three transects. These mean values were then used in Spearman's rank correlation tests to determine whether there were any correlations between the benthic group cover and year. Only sites that had at least five or more years of observation were considered for this study. Year 2001 was also not considered as only four back- and two fore-reef sites were surveyed, and therefore the data for this year were not representative of the whole island.

Temporal changes in total stony coral percent cover were also studied. A long-term mean for each site was calculated by averaging the annual percent cover of total live stony coral for the 13-year period for each site. This long-term mean for each site was subtracted from their respective annual means and the resulting values were used to create anomaly plots. These plots showed the mean annual deviations in total live stony coral cover for the 1998 to 2010 period. Two sets of anomaly plots were created, one from summer data and one from winter data.

### Land-based stressors—relationship with benthic cover trends

Temporal trends in land-based stressors were first examined by plotting total human population, total land area under agriculture, total number of tourists and total rainfall per year for the whole island against time. A Z-score was used on the $y$-axis to facilitate comparisons between stressors, which had widely different ranges.

To investigate the effects of the land-based stressors on the benthic communities, we used a combination of non-Metric Multidimensional Scaling analyses (nMDS) on Bray-Curtis distances (*Borcard, Gillet & Legendre, 2011*) and the 'envfit' function of the package vegan in R (*Oksanen et al., 2015*). The mean percent cover per benthic group per year was first calculated using data from both summer and winter; the seasonal data were combined since we only had one set of annual data for the land-based stressors (no seasonal data) to which the benthic data were tested with. The mean percent cover values were then used in nMDS to generate ordination plots for each site. The ordination plots indicate the
scale of changes in community structure at each site. Then, using the 'envfit' function, data from the four stressors were fitted onto these ordinations to determine which ones had a significant correlation with the placement of benthic community observations in ordination space. The 'envfit' function considered the effects of each stressor individually.

## RESULTS

### Changes in benthic cover
#### Individual sites
Stony corals were the most abundant group at most sites. The back-reef sites had in general more *Acropora* corals while the fore-reef sites had more of the non-*Acropora* corals (Fig. 2). All sites had rubble and/or rocks (non-living category). The fore-reefs had in general more rocks. Turf algae and macroalgae were present at most sites albeit in varying amounts. Coralline algae were rarely detected. All sites had some dead corals; back-reef sites tended to have more dead corals than the fore-reef sites. Very little of other living organisms were recorded at the study sites.

Changes in benthic cover varied a lot among sites (Table 1). The 'X' was used to indicate benthic groups not observed during the survey. The *Acropora* coral groups experienced a decrease in cover over time. Although some non-*Acropora* coral groups also showed a significant decrease, a few groups increased in abundance. In particular, non-*Acropora* encrusting increased in cover at four back-reefs sites. Non-*Acropora* foliaceous, massive and submassive also increased at one site each.

Other benthic groups studied showed variable responses (Table 1). The cover of dead corals increased at four sites and decreased at one site. The cover of rubble increased at one site and decreased at three sites. Turf algae cover increased at one and three sites. The cover of macroalgae increased at three sites and decreased at two sites. Coralline algae increased at two sites and soft corals at one site. Sand increased at two sites and decreased at three sites. *Millepora* and sponges respectively decreased at one site and increased at one site. The group 'other live' decreased at two sites. No clear seasonal signal was detected except for turf algae and macroalgae, which appeared to be more abundant in summer than in winter.

#### Anomaly plots—stony corals
The summer anomaly plots showed that a large number of sites had a mean annual cover above their respective long-term mean (more circles with warm colors) until 2005, after which this trend was reversed; most sites afterwards had a mean annual cover below their respective long-term mean (more circles with cold colors; Fig. 3). Not a single site in 2009 and 2010 had a mean annual cover above their respective long-term mean. A similar trend was observed from the winter anomaly plots, although not as clear as those from the summer data (Fig. 4).

### Island-wide trends for land-based stressors
When local stressor trends for the whole island were examined, there were clear increases in human population and the tourism index over the study time (Fig. 5). There was a

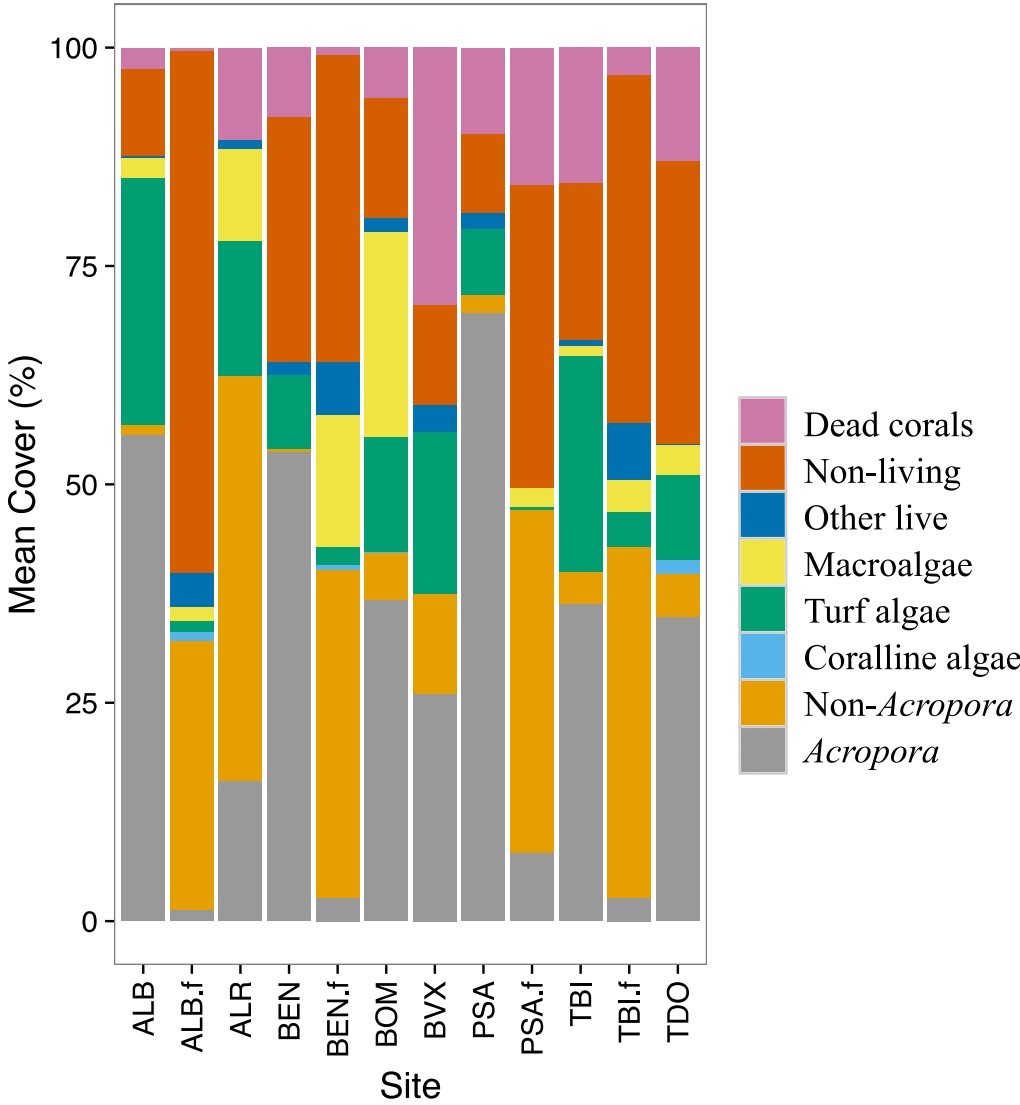

**Figure 2  Benthic cover by site for 1999.** The benthic cover data (23 categories) were aggregated into eight groups to facilitate data visualization. The '.f' indicate a fore-reef site. See methods for full site names.

general decline in land area under agriculture from 2001 and 2008, after which more land was cultivated. The big dip in cultivated land area in 1999 coincided with the big dip in annual rainfall, the year with the lowest rainfall during the study period. There was more rain after 1999, but it fluctuated quite a bit over time.

## Land-based stressors—relationship with benthic cover

Significant effects of human population were detected at 11 of the 15 study sites (Table 2). Tourism and agriculture had significant effects at five and four study sites respectively. No significant effects of rainfall were detected at any of the study sites. There were insufficient data to carry out this test at Belle Mare (BME). The 'X' in the table indicates that there
**Table 1 Temporal trends in benthic cover at individual sites. Spearman rank correlation results.** Only significant results are shown ($p < 0.05$). The 'X' indicates that benthic group was not observed. Only sites that had at least five years of observation were used in this test. See methods for full site names.

| Reef (Back or Fore) | Sites | Season (S or W) | n | Acropora branching (ab) | Acropora digitate (ad) | Acropora encrusting (ae) | Acropora submassive (as) | Acropora tabulate (at) | non-Acropora branching (nb) | non-Acropora encrusting (ne) | non-Acropora foliaceous (nf) | non-Acropora massive (nm) | non-Acropora mushroom (nu) | non-Acropora submassive (ns) |
|---|---|---|---|---|---|---|---|---|---|---|---|---|---|---|
| B | ALB | W | 12 | −0.87 | X | X | X | | | | −0.58 | | | −0.80 |
| B | ALR | S | 7 | | X | | X | X | | | −0.96 | | | |
| B | ALR | W | 6 | | X | | X | X | | | | | | |
| B | BDT | W | 6 | | X | X | X | | X | +0.87 | | | | |
| B | BEN | S | 8 | | −0.76 | X | X | −0.91 | | | X | | | |
| B | BEN | W | 7 | | | X | X | | −0.80 | | X | | | |
| B | BME | W | 6 | | X | X | X | | | | X | X | X | X |
| B | BOM | S | 8 | −0.71 | X | X | X | | | | | X | | |
| B | BOM | W | 7 | | X | X | X | | | | | X | | |
| B | BVX | S | 8 | | | X | | | | +0.87 | +0.70 | | | |
| B | BVX | W | 8 | −0.88 | | X | | −0.73 | −0.85 | | +0.95 | | | |
| B | PDO | S | 6 | X | X | X | X | | X | | −0.93 | −0.84 | | −0.82 |
| B | PSA | S | 8 | −0.78 | X | X | | X | | X | X | | | |
| B | PSA | W | 5 | | X | X | | X | | X | X | | | |
| B | TBI | S | 9 | | | X | X | −0.78 | X | +0.78 | | | | |
| B | TBI | W | 5 | | | X | X | | X | | | | | −0.86 |
| B | TDO | S | 9 | | | X | X | −0.73 | −0.73 | +0.83 | | +0.67 | | +0.82 |
| B | TDO | W | 6 | | | X | X | | | | | | | |
| F | ALB | W | 10 | | | | | X | | −0.76 | | | X | |
| F | BEN | S | 5 | | | | | X | | | | | X | |
| F | BEN | W | 7 | | −0.97 | | | X | −0.80 | | | −0.82 | X | |
| F | PSA | S | 8 | | | | −0.76 | X | | | | | | −0.80 |
| F | TBI | S | 6 | −0.84 | | X | | | −0.84 | | | | X | −0.86 |
| F | TBI | W | 6 | | | X | | | | | | | X | |

| Reef (Back or Fore) | Sites | Season (S or W) | n | Turf algae (ta) | Macroalgae (ma) | Coralline algae (ca) | Millepora spp. (mi) | Sponges (sp) | Soft corals (sc) | Zoanthids (zo) | Dead corals (de) | Rubble (ru) | Sand (sa) | Rock (ro) | Other (ot) |
|---|---|---|---|---|---|---|---|---|---|---|---|---|---|---|---|
| B | ALB | W | 12 | −0.81 | | | X | | | X | | +0.83 | | | |
| B | ALR | S | 7 | +0.79 | | | | | | X | +0.85 | | | X | |
| B | ALR | W | 6 | | | | | | | X | | | | X | |
| B | BDT | W | 6 | | | | X | | X | | | −0.94 | | X | X |
| B | BEN | S | 8 | | | | X | | | X | +0.83 | | | | X |
| B | BEN | W | 7 | −0.93 | | +0.75 | X | | X | X | +0.82 | | +0.86 | | −0.80 |
| B | BME | W | 6 | | +0.94 | X | X | X | X | X | | −0.83 | | | X |
| B | BOM | S | 8 | X | | | X | X | X | | | | −0.79 | X | |
| B | BOM | W | 7 | | | | X | X | X | | | | | X | |
| B | BVX | S | 8 | | | | | X | X | | | | | X | |
| B | BVX | W | 8 | | | | | X | X | | +0.71 | | | X | |

| Reef (Back or Fore) | Sites | Season (S or W) | n | Turf algae (ta) | Macroalgae (ma) | Coralline algae (ca) | Millepora spp. (mi) | Sponges (sp) | Soft corals (sc) | Zoanthids (zo) | Dead corals (de) | Rubble (ru) | Sand (sa) | Rock (ro) | Other (ot) |
|---|---|---|---|---|---|---|---|---|---|---|---|---|---|---|---|
| B | PDO | S | 6 | | +0.82 | | X | | X | X | | | | X | X |
| B | PSA | S | 8 | | +0.91 | | X | | | X | −0.90 | +0.95 | | | |
| B | PSA | W | 5 | | | | X | | | X | | +0.90 | −0.90 | | |
| B | TBI | S | 9 | | | X | | X | X | X | +0.78 | | −0.68 | | X |
| B | TBI | W | 5 | −0.94 | −0.93 | X | | X | X | X | | | | | X |
| B | TDO | S | 9 | | | | X | | | | | −0.82 | | | |
| B | TDO | W | 6 | | | | X | | | | | −0.93 | | | |
| F | ALB | W | 10 | | −0.78 | −0.70 | | | +0.84 | | | | | | |
| F | BEN | S | 5 | | | | X | X | X | | | | | X | |
| F | BEN | W | 7 | | | | X | X | X | | | | | X | |
| F | PSA | S | 8 | | | | | X | | X | | | | | |
| F | TBI | S | 6 | | | | X | | | X | | | | | |
| F | TBI | W | 6 | | | +0.97 | X | −0.83 | | X | | | +0.81 | | −0.82 |

were either only a few years or no tourism data in the watersheds linked to these study sites. Therefore, it was not possible to test for the effects of tourism at these sites.

## DISCUSSION

Some significant changes in benthic cover occurred around the island of Mauritius during the period of 1998 to 2010. Among the stony corals, *Acropora* coral groups appeared to have been the most impacted, decreasing in abundance at many sites. While the non-*Acropora* coral groups also decreased in abundance at many sites, some groups however experienced an increase over time. The increase in abundance of dead corals and rubble at some sites also supported the observations of stony coral decline during the study period. Additionally, the decline in stony corals appeared to be more important in second half of the study period for all sites. There was little change in cover for the other benthic groups, some of which were quite rare. Our study also suggests that humans are an important factor contributing to the demise of coral reefs around the island.

The decline of *Acropora* corals around the island was expected as *Acropora* corals have been reported to be most sensitive to natural and anthropogenic stressors (*Marshall & Baird, 2000*; *Loya et al., 2001*; *Rachello-Dolmen & Cleary, 2007*; *Cleary et al., 2008*; *Côté & Darling, 2010*). In contrast, the non-*Acropora* coral groups, particularly ones with encrusting morphologies, appeared to be more successful competitors on the reefs. Their morphology might be responsible for a higher mass transfer rate for gases and nutrients (*Patterson, 1992*), which in turn could lead to faster removal of harmful superoxides and toxic radicals produced during stressful events (*Lesser, 1997*). A study done in Japan also reported higher survival of corals with massive and encrusting morphologies after a bleaching event (*Loya et al., 2001*). This increase of encrusting corals coupled with the loss of branching, digitate and tabulate *Acropora* corals also suggest that the coral reefs around Mauritius have flattened and experienced a decline in habitat complexity. Similar flattening of reefs has been observed in the Caribbean (*Alvarez-Filip et al., 2009*).

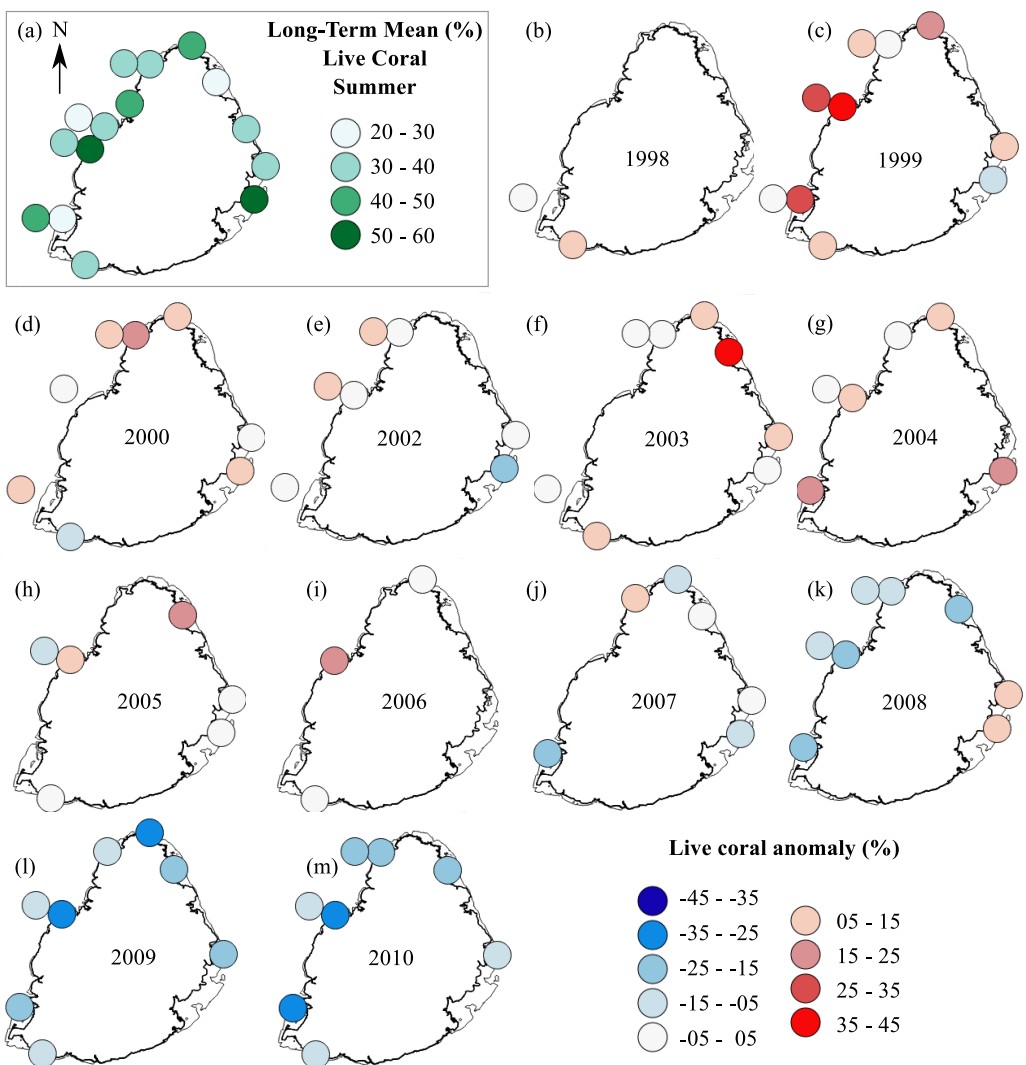

**Figure 3 Anomaly plots for stony coral cover—Summer data.** (A) Long-term means for each site for the period of 1998–2010; year 2001 was not included as very few sites were surveyed during that year. Anomaly plot for (B) 1998, (C) 1999, (D) 2000, (E) 2002, (F) 2003, (G) 2004, (H) 2005, (I) 2006, (J) 2007, (K) 2008, (L) 2009 and (M) 2010. Warm colors indicate coral cover was above long-term mean and cold colors indicate that coral cover was below long-term mean.

Interestingly, all study sites appeared to have experienced a more significant decline in stony coral cover during the second half of the study period, i.e., around 2005 and onwards. These observations suggest that a global rather than a local factor was involved. The 50-km Degree Heating Weeks (DHW) data from Coral Reef Watch NOAA website for 2005 showed that the reefs around Mauritius experienced severe heat stress (4 °C-weeks and above) for three consecutive summer months (ending 29 March, 30 April and 31 May; Table S3). Therefore, temperature was a most likely stressor contributing to impact severely the corals during 2005, which led to subsequent declines in cover. The temperature stress from 2005 was so severe that the corals had not recovered by 2010.

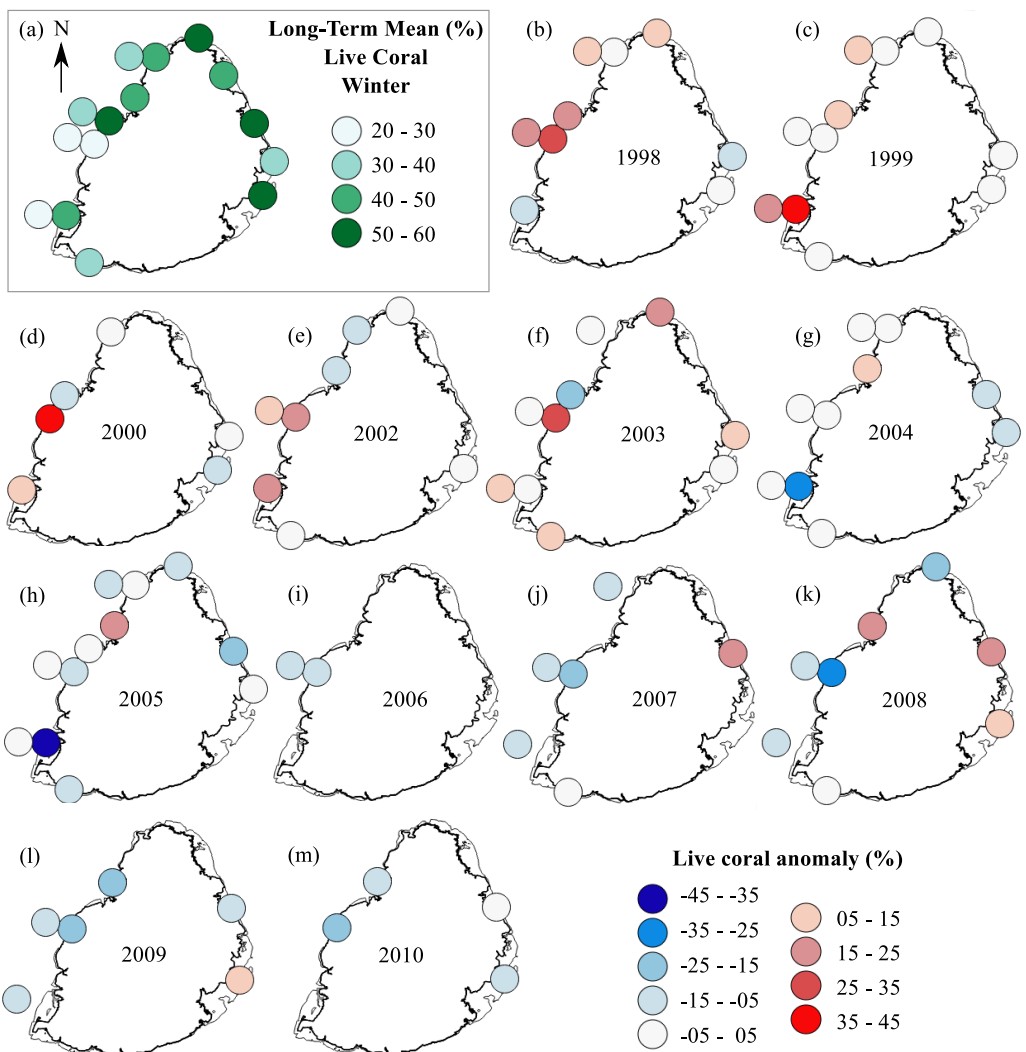

**Figure 4** **Anomaly plots for stony coral cover—Winter data.** (A) Long-term means for each site for the period of 1998–2010; year 2001 was not included as very few sites were surveyed during that year. Anomaly plot for (B) 1998, (C) 1999, (D) 2000, (E) 2002, (F) 2003, (G) 2004, (H) 2005, (I) 2006, (J) 2007, (K) 2008, (L) 2009 and (M) 2010. Warm colors indicate coral cover was above long-term mean and cold colors indicate that coral cover was below long-term mean.

The cover of other benthic groups such as macroalgae and turf algae did not change much at most of the study sites, but they did appear to grow more in summer. These observations suggest that there was in general a good top-down control by herbivores at these study sites. Herbivore biomass is known to play an important role in coral reef dynamics and in mediating reef regime-shifts (*Jouffray et al., 2015*). As observed on the Great Barrier Reef (*De'ath, Lough & Fabricius, 2009*), herbivorous fish on the reefs of Mauritius also face less significant fishing pressure as artisanal fisheries target mainly piscivorous fishes.

Coralline algae is important for coral larval settlement (*Harrington et al., 2004*) and their cover did not change much at most sites; they only increased at two sites. *Millepora* spp,

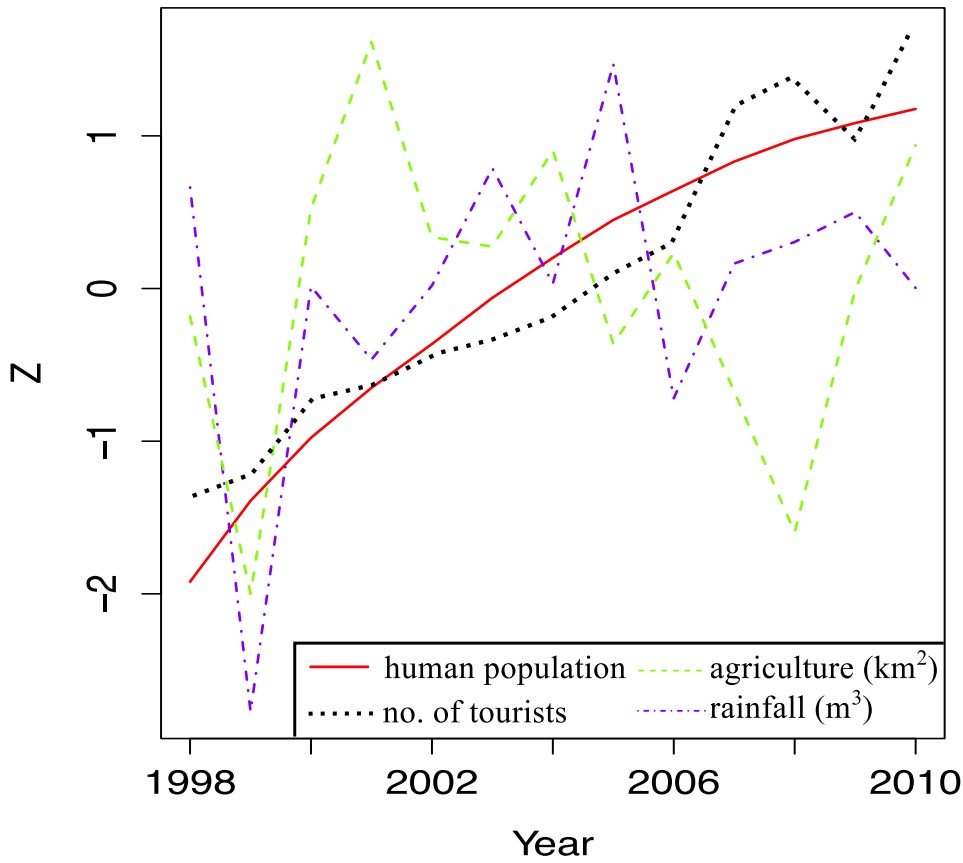

**Figure 5 Trends in land-based stressors for 1998–2010.** The total human population (total number of local people), total number of tourists, total surface area under agriculture and the total rainfall for each year are shown.

sponges, soft corals, zooanthids and other live group were either quite rare at most sites and/or did not appear to experience much change during the study. The lumping of species into larger groups could, however, have obscured temporal trends for individual species. For these rare groups, more targeted approaches are needed to study them.

Human population was the land-based stressor that appeared to impact the most sites around Mauritius. High human densities have previously been reported to impact negatively coral reef communities (*Mora, 2008*; *Sandin et al., 2008*; *Smith et al., 2016*; *Crane et al., 2017*). However, as also argued, it is not the number of people, but rather their activities that affect the coral reefs (*Sanderson et al., 2002*; *Mora, 2008*). Tourism was the second most common stressor to impact the reefs. Therefore, it appeared that the activities of the growing local population and the tourism industry have contributed the most to affect the coral reefs. The fore-reef sites located on the west coast also appeared to be primarily impacted by human population. These observations suggest that fore-reefs may not be far enough from land to be isolated from the impacts of the local human population.

**Table 2  Land-based stressors and their relationship with benthic cover.** Asterisks indicate that relationships were significant ($p < 0.05$). The 'X' indicates that the tourism data were insufficient for that site.

| Sites | N | Reef | 2-D Stress | Population | Tourism | Agriculture | Rainfall |
|-------|---|------|------------|------------|---------|-------------|----------|
| ALB | 12 | Back | 0.03 | * | X | * | |
| ALR | 11 | Back | 0.09 | * | * | | |
| BDT | 8 | Back | 0.06 | | | | |
| BEN | 11 | Back | 0.06 | * | * | * | |
| BME | 7 | Back | Insufficient data | | | | |
| BOM | 12 | Back | 0.20 | | X | | |
| BVX | 11 | Back | 0.09 | * | X | | |
| PDO | 8 | Back | 0.04 | * | * | * | |
| PSA | 11 | Back | 0.07 | * | | * | |
| TBI | 11 | Back | 0.11 | * | * | | |
| TDO | 11 | Back | 0.10 | * | X | | |
| ALB | 11 | Fore | 0.09 | * | X | | |
| BEN | 10 | Fore | 0.11 | | | | |
| PSA | 10 | Fore | 0.10 | * | | | |
| TBI | 10 | Fore | 0.09 | * | * | | |
| TOTAL | | | | 11 | 5 | 4 | 0 |

However, we had data for only four fore-reef sites all located on the west coast of the island, and this small sample size may not be adequate to draw broad conclusions.

Agriculture appeared to affect only four study sites, however total land area under agriculture was used as a proxy for its effects and may not be representative of its negative impacts, e.g., the excessive use fertilizers and their detrimental effects are well documented (*Markey et al., 2007*; *Brodie et al., 2012*). Similarly, rainfall did not appear to affect the coral reefs; the negative impacts of runoff bringing pollutants to lagoons are also well documented (*Fabricius, 2005*; *Brodie et al., 2012*). Therefore, investigating the impacts of specific human activities would provide greater insights on the root causes of reef degradation around Mauritius.

The coral reefs as we know them are changing rapidly and will continue to degrade under the pressure of multiple natural and anthropogenic stressors. How coral reefs are going to change and how to protect them so they maintain important ecosystem functions for future generations remain a challenge. Our results showed that *Acropora* corals have been declining and suggest that corals reefs are flattening. Moreover, our results also suggest that humans are a contributing factor to coral reef degradation and loss. As ocean and atmospheric warming continue, reducing the impacts of human activities remains very important. Other studies focusing on biological traits that convey resilience in the face of multiple stressors are needed to better understand how benthic communities are going to change in the future.

## ACKNOWLEDGEMENTS

We would like to thank the Albion Fisheries Research Centre, Ministry of Fisheries, Government of Mauritius for providing the long-term benthic community dataset without which this work would not have been possible. Jennifer Elliott expresses her deepest gratitude to the members of her advisory committee, Peter Edmunds, Tarik Gouhier, Brian Helmuth and Steve Vollmer for their thoughtful inputs during the execution of this work. This is contribution number 384 from the Marine Science Center at Northeastern University.

### Funding

Support for this work came from a Northeastern University Dissertation Completion Fellowship (Jennifer A. Elliott), start-up support (Mark R. Patterson), and NSF Award # 1412462 (Mark R. Patterson). The funders had no role in study design, data collection and analysis, decision to publish, or preparation of the manuscript.

### Grant Disclosures

The following grant information was disclosed by the authors:
Northeastern University Dissertation Completion Fellowship.
NSF Award: #1412462.

### Competing Interests

The authors declare there are no competing interests.

### Author Contributions

- Jennifer A. Elliott conceived and designed the experiments, performed the experiments, analyzed the data, prepared figures and/or tables, authored or reviewed drafts of the paper, approved the final draft.
- Mark R. Patterson conceived and designed the experiments, contributed reagents/materials/analysis tools, authored or reviewed drafts of the paper, approved the final draft.
- Caroline G. Staub analyzed the data, authored or reviewed drafts of the paper, approved the final draft.
- Meera Koonjul contributed reagents/materials/analysis tools, authored or reviewed drafts of the paper, approved the final draft.
- Stephen M. Elliott conceived and designed the experiments, analyzed the data, authored or reviewed drafts of the paper, approved the final draft.

### Data Availability

Raw data are provided in the Supplemental Files.

## Supplemental Information

Supplemental information for this article can be found online at http://dx.doi.org/10.7717/peerj.6014#supplemental-information.

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
