# Peer review of "Decline in coral cover and flattening of the reefs around Mauritius (1998–2010)"

_PeerJ, doi:10.7717/peerj.6014_

## Round 0.1 · original submission · Major Revisions

Please respond appropriately to all the points made in the 3 detailed reviews

Please make sure that you focus on the hypotheses for which the data are suited. This weakness was noted by more than one reviewer

Reviewer 1 ·

Basic reporting

The basic reporting is sufficient. There is redundancy in the text and the use of references (e.g., clumps of references that travel together across sections), which is stylistically distracting and not as precise as possible.

Structurally, there are a number of hypotheses that seem ambitious and wide-ranging.

Experimental design

The core of this report is to present data of the benthic structure from a ~12-year monitoring program. The data themselves are valuable and interesting to explore. The hypotheses, as noted above, are wide-ranging and extend a bit beyond the statistical power within the data. With 11 sites that vary in habitat (though the map has 15 sites[?]), there is limited statistical power to address correlational patterns. This is especially true given the high variability implicit to such observational data.

So while the data are valuable, the hypotheses are exploratory and perhaps too broad for the data presented.

Validity of the findings

The conclusions regarding the observed patterns in the benthic composition are robust. The tests of candidate hypotheses are prone to Type I error (i.e., spurious relationships due to high variability, poor fit to model assumptions, and multiple comparisons). For example, a jack knife analysis would be interesting -- try omitting data from one site and see if the statistical fits remain consistently.

So what are presented as tests of hypotheses may be better placed into the realm of 'speculation' and exploratory analysis.

Additional comments

The manuscript is long and redundant. There are many hypotheses introduced that extend well beyond the strength of the data (e.g., attributions of cause-and-effect). Further, it is unclear what the statistical power of the benthic assessments are at their foundation. How many points were used in the line-point-intercept analysis? Such information on site-specific sampling effort is critical in order to interpret observed patterns. Of course, a sincere strength of this study is the time-series element, where statistical strength is included through application of a pseudo-regression based approach (connecting data across sampling intervals and assessing temporal trends).

The introduction and analyses are something of a 'kitchen-sink', where the study is introduced with a broad spectrum of putative environmental relationships and the analyses force the data to explore each hypotheses. I believe that the data are being stretched further than they can perform here, and the study would be better suited as a presentation of (really strong!) temporal trends with more targeted analyses and interpretations.

·

Basic reporting

Writing and organization are generally fine, I have some comments about specific lines, submitted under "other" below.

Experimental design

This is a study which analyzes data from a monitoring program. It appears that the monitoring program was designed and implemented long before the analysis was done, by different people than those that analyzed the data, and the monitoring was not designed by those that analyzed the data and thus not designed specifically to make these particular analyses possible or robust. (this is a common situation) As a result, the design has a number of flaws that are common in monitoring programs, but which if the programs were designed for statistical analyses, would have been designed and carried out differently. That leaves a variety of weaknesses that the analysis can’t completely compensate for, such as not randomizing locations, having only four reef slope sites, and so on.
The analysis appears to be sophisticated, but suffers from a typical problem of modern, high-tech, statistical analyses, in that simple natural history observations preferably with some data recorded, could have documented the effects of bleaching and a cyclone directly. Direct measurement of variables is almost always superior to indirect inference and guessing based on fancy statistics.
The monitoring program design was pretty good for detecting change, since sites were marked. Apples need to be compared with apples, not oranges, and if the sites change from year to year the measurement of temporal change is contaminated with spatial effects. Figure 2 does show which sites were done which years. 2001 has a particularly low number of sites and probably should be excluded from temporal analysis. Figure 2 shows that clearly the decrease in coral cover in the average wasn’t due to changes in which sites were recorded, by the end all sites were in negative territory. So these problems may (may) be more theoretical than practical. (but coral cover is only one part of the analysis) But it would be good to have a way to estimate how much spatial differences contribute separate from time to the averages. This is a very common problem.

Validity of the findings

Since sites were not chosen randomly, they are not necessarily representative of the various areas and thus concluding which variables controlled the variables being mentioned, could be influenced by site selection bias. Could be the results are fine. Bias in choosing sites is common, people want to monitor places that have good coral. Unfortunately, the sites and collection of data can't be changed.

Additional comments

Line by line comments:
Line 42 the statistical methods aren’t necessary in the abstract unless they are the focus of the paper. They don’t particularly hurt the abstract, just aren’t necessary.
Line 54 “however” doesn’t seem necessary
Line 75 wouldn’t you say that corals are very near the limits of their thermal tolerance during the summer? About 1 degree above mean max temp and they start bleach, 2 degrees and they start to die. They aren’t quite at threshold most years even at peak summer temperatures.
Line 86 driven to extinction. Some have certainly been driven to local, economic and/or ecological extinction, but few in the marine world have been driven to global extinction.
Line 89 in the Caribbean
Line 115 a new study has also shown that light adds to heat stress in producing bleaching: Coelho, V., Fenner, D., Caruso, C., Bayles, B., Huang, Y., Birkeland, C. 2017. Shading as a mitigation tool for coral bleaching in three common Indo-Pacific species. Journal of Experimental Marine Biology and Ecology 497: 152-163.
Line 120 regularly exposure
Line 140 contribute to the impact on
Line 167 that is a small number of sites, particularly on the forereef, for such a large island.
Line 167 how were the sites chosen? They appear well spaced out along the coastline. The back reef sites are located in a shallow “lagoon”, is the lagoon mostly sandy? It is very wide in places, so how were sites chosen within that large area? On the reef slopes, sites were located at a range of depths. The depth could affect the reef community, how was the effects of depth separated from the other effects of environmental variables?
Line 172 one of the back reef site
Line 187 “permanent” is a popular word for a marked or staked transect. However, little on earth if anything is permanent (since earth probably will last for another roughly 4 billion years) and especially on coral reefs where a cyclone can rip stakes out. Marked or staked is more descriptive. Staked or marked transects are likely to be better than transects relocated in other ways. I presume the stake or mark was at the start of the 3 transects, and there were no further stakes or marks along the transects, is that so? I presume that the transects were laid on depth contours on the forereef, you should state that. At the backreef sites, how were the tape locations determined other than the stake at the beginning of the transect? By compass bearing? I assume they were on nearly flat substrate and the transect didn’t follow a depth contour, is that the case?
Line 209 Fungiid
Line 210 organisms
Line 267 need to define GHRSST
Line 272 suggest “these” instead of “they”
Line 274 ENSO certainly causes variation on an annual scale. As far as I know, it is largely independent and on top of climate change.
Line 298 I think it would be worth showing (perhaps in a stacked bar graph) the cover for each site for one year, and the benthic trends of the means over years so the reader has some idea of the composition and trends of the categories.
Lines 304-305 how much variation was there in how many sites were surveyed in each year? If the differences between sites were large (as they often are) then changes in which sites were surveyed can produce changes in the mean that are produced by spatial changes instead of the changes over time that you wish to measure. If the number of sites and which sites change from year to year, how does the analysis remove the spatial contribution from the mean to reflect only changes in time? (Figure 3 answers my question)
Line 326 not sure what you mean by spatially averaged. Can that be explained?
Line 328 are these spatial averages for the forereef sites only, for backreef sites only, or for forereef and backreef sites? Were the spatial averages done in two spatial dimensions or one? Was it assumed that the reef area around a site had the same cover as the site? Was cover interpolated between sites? Any other assumptions?
Line 436 rubble
Line 557 in moving water (in still water they are presumably similar)
Line 568 other studies in the Pacific implies that this study was in the Pacific. Just remove “other” or change it to Indo-Pacific.
Line 585 are there any records of bleaching in Mauritius? SST being the only variable correlating with the decline of coral certainly suggests bleaching might be the cause but it would sure be good to back that up with some information about actual bleaching events. If there were no bleaching events there during this period, then some other process caused by SSTs probably caused the loss of coral cover. Fancy statistics and assumptions like this are no substitute for in the water direct observations and recording of the effects of disturbances. If you don’t have natural history observations, you can easily make mistakes with assumptions of causality.
Line 589 the logic behind this statement isn’t obvious.
Line 590 often causes(?)
Line 592 again, direct observation of cyclone damage is a far superior way to study the effects of cyclones than fancy statistics.
Line 604 as from?
Line 605 the trend with community structure was similar to that of many of the benthic groups studied? But isn’t the community structure composed of benthic groups?
Line 629 I don’t see why you say “however”
639 different to suggest different from
648 the resistance index
658 the differences between the width of the back reef at different sites is far greater than the distance between the back reef sites and fore reef sites. So if the distance of the backreef sites from shore doesn’t affect the coral community, it is unlikely that distance alone is a good explanatory variable for the difference between backreef and forereef sites. Do people walk out on the backreef for fishing and/or gleaning, or is fishing pressure much heavier on the backreef than forereef?
666 biodiversity correlates positively
673 coral reef
740 Porites astreoides needs to be italicized

Figure 3a you mean that a count of the human population in Mauritius is done every year, and in 2001 there was a big drop in population which was restored the next year? Did a whole bunch of people move off the island and then return a year later? Certainly a whole bunch of people didn’t die and then get reborn the next year. This data looks suspect. Population counts in the US are only done every 10 years, and are quite expensive since it is a huge task.
Figure 3b the dotted lines for cyclonic index and ENSO index are not the same as in the graph, and without blowing it up greatly are very hard to distinguish. If they were the same as on the graph they would be much easier to distinguish. I’d suggest thickening the lines as well.

Reviewer 3 ·

Basic reporting

The authors use clear, professional English language throughout the manuscript. The main point of the title (i.e., fore-reefs as refugia) seems in direct contrast to the first statement about results in the abstract that “There was in general a significant decrease in live stony coral cover and a significant increase in dead coral and/or rubble cover island-wide for both the back- and fore-reefs.” (L43-45). This statement is supported by the island-wide results for benthic cover that report significant declines in more stony coral groups for fore-reefs (7 groups) than for back-reefs (5) (L431-440). In general, the study design, data analysis, and results do not support the conclusion as presented in the title. Suggest an alternative such as “Changes in coral reef cover and community structure for Mauritius,1998-2010, from anthropogenic and environmental stressors”. The Introduction provides background for the context of the paper but there are statements that either cite a reference inappropriately (e.g., L73 citation (De’ath) does not support statement) or do not list a reference (L69-70 citation(s) needed). A few specific recommendations for text improvements include L50 edit “population” to “human population”, L101 edit “reef organisms” to “coral species” as they are the focus of the paragraph, not all reef organisms, L120 edit “regularly” to “regular”, and L139 needs a citation for statement about less fisherman, probably due to overfishing. The structure of the paper conforms to PeerJ and discipline standards. The Figures and Tables are relevant and professional but required edits. The Figure 2 legend station name abbreviations don’t match those listed in the raw data table. Need to include a description of what “anomaly” represents here. For example, the anomaly is a percentage above or below the long term coral cover mean for a given year. The legend for the anomalies includes a range of values (-61 to -50 and -50 to -40) that doesn’t appear in the data for any year. If that is the case, remove those from the legend and recolor the blue dots with a stronger gradient so that the differences in values is easier to interpret. Table S1. You don’t need to include the fore reef sites (e.g., NAME.f) in the list since they just present entries with duplicate values. Your Methods describes the site names and identifies that some sites have both back and fore reef sites. Table S3. For what time period is the Evenness index calculated?

Experimental design

The primary research is original and within the scope of PeerJ. The research question is fairly well defined and meaningful for the field. The Introduction sufficiently details how the work contributes to the discipline. The Investigation was performed to a reasonable technical standard but there were a number of issues. A primary concern is drawing inferences for the island-wide trends in benthic cover drawn from an overall low sample size, the unbalanced sampling design between fore-reef and back-reef sites, and uneven gaps in samples for time series. Your study includes only 4 fore reef sites from the leeward side of the island and compares these to 11 back reef sites from both the leeward and the windward sides. The lack of any windward fore reef sites undermines the inferences that can be made regarding the back reef and fore reef for the entire island domain. Furthermore, using a visual observation of the long-term coral cover means and the coral cover anomalies through time for the four locations that had both back reef and fore reef sites in Figure 2, the fore reef sites had either lower or similar coral cover to the adjacent back reef sites and for a number of years had instances of lower coral cover anomalies than the adjacent back reef sites (e.g., 1998-2001, 2003-2004). Figure 2 also highlights gaps in the time series of data for particular locations with approximately 19% of your samples missing from all possible site x time combinations and only one site (most southerly) having samples for the entire time period. Many of these issues are addressed for your community structure analyses but basically ignored for the trends in benthic cover analysis.

I had a concern about the coarse resolution of your global stressors, in particular SST derived from satellite data as a proxy for subsurface water temperatures on the nearshore benthic ecosystems. The accuracy of satellite-based SST in shallow, near-shore environments may be adversely affected at sub-regional scales by land adjacency as well as localized optical and oceanographic factors, which introduce bias into analyses (Castillo and Lima (2010) Limnology and Oceanography: Methods 8: 107–117, Leichter et al. (2006) Journal of Marine Research 64: 563–588, McClanahan et al (2007) Ecological Monographs 77:503-525. For example, Smale and Wernberg (2009) found that SSTs were significantly different from benthic water temperatures (usually 1 to 2°C higher), and did not adequately detect ecologically important small-scale variability or provide reliable information on temperature extremes. The use of a single SST per year seems too simplistic since coral bleaching dynamics are driven by extreme high (or low) values. Given that “Remotely sensed SST data were used since observed data were sparse and not representative.” (L265-266), I encourage you to demonstrate a statistical relationship between the remotely sensed data that you used for your study and the available observed seawater temperature data. This would help justify the use of the satellite data. Since the crux of your time series changes seem to rely on the shift in coral cover around 2004-2005, I suggest that you include one or a few of the available SST satellite coral bleaching indices (such as degree heating weeks) that may be more appropriate to use than just SST to predict bleaching occurrence.

The description of study sites does not include mention of the seasons, months, or dates when sites were surveyed. Considering the temporal aspect of this study, it is important to describe when the sites were surveyed within each year as differences in biological observations may reflect season differences (especially for algal groups, Williams (1993) JEMBE 167: 261-275), thus confounding any inferences that you draw regarding inter-annual differences.

Are the transects at sites truly permanent? (L186-187) In other words, is the exact same location (e.g., permanent stakes in the reef) resurveyed during each sampling occasion or just the general vicinity? If the former is the case, you’ll have issues with the independence of your samples across time and your statistics should incorporate the structure of your data (i.e., through repeated measures or mixed modeling approaches).
Did the line transect field method include a measure of benthic complexity like rugosity? This would allow you to address a loss of structural complexity with declines in coral cover. Although not ideal, if you didn’t collect rugosity measures during the study, you might consider performing an ancillary study to collect a series of rugosity measures for the various coral categories included in this study. This would allow you to estimate a mean rugosity for each category and then calculate a weighted-mean rugosity from each line transect. An examination of changes in rugosity would provide evidence regarding the outcomes of your observation that some encrusting or massive groups increased while the Acroporids decreased (i.e., homogenization of habitat complexity).

I don’t agree with your definition of a global stressor. Like your definition of a local stressor, a global stressor has different levels of impact at different sites but the source of the stressor is not local (e.g., SST or cyclones generated from global and regional processes of the coupled ocean-atmosphere system can have variable impacts depending upon site location). The source of the local stressor is local (e.g., rainfall as a proxy for run-off from an adjacent watershed differs between watersheds).

Methods for the analysis of trends are not described in sufficient detail. How were missing data years dealt with in the calculation of a long-term mean? When were stations surveyed during each year (i.e., is season or month a missing factor in your analysis or was it controlled for by sampling during the same month/season each year)?

Methods for analysis of community structure and stressors finds no apparent change in community structure for fore reefs in the analysis yet these sites declined significantly in Acropora and non-Acropora coral cover. I suggest that the analyses should be modified by lumping certain categories (e.g., by coral structure or taxonomy) so that you do not sacrifice data as described in your exclusion of rare categories (L352-353).

Methods for benthic community resistance and recovery. Need to define “Biodiversity” as a metric (L384). For the resistance index, you need to clarify what constitutes the “21-group benthic data for all years for which data were available” (L387-388). The logical structure of this index is a bit confusing since you’re assuming equal strength of stressors at each site but have statistically shown a correlation with population and back-reef communities. Doesn’t that finding undermine the utility of this approach? Delete the section describing a test of “wind” on the resistance index since the test was not run. For the test of biodiversity on resistance and recovery, please provide evidence that the number of coral colony morphology categories is a suitable proxy for coral species diversity. If no evidence exists, please delete this analysis and accompanying results from the manuscript. Given the relatively short time between the observed large decline in coral cover around 2004-5 and the end of the time series (2010), it doesn’t seem reasonable to assume that recovery would occur on a 1-5 year time scale for corals. I’m not sure there is much utility for this analysis given this constraint. It also seems out of place to present the resistance index stratified by sites exposed to or sheltered from wind since this variable wasn’t part of any of your other analyses due to the incompleteness of the sampling design (i.e., insufficient windward/leeward replication for fore reef).

Validity of the findings

The general conclusions drawn about the study of fore-reefs as a refugia are not well supported in the study results. I suggest a reframing of the study that does not draw this conclusion as the central focus of the paper but instead promotes when/what/how of the large decline in corals during the time series. There should be additional analysis to examine the effects of the 2004-2005 change point in the decline of corals regarding SST anomaly indices, not just a single SST value per year. This could also include some reconsideration of categories to include in your multivariate analyses from lumping by taxa and/or morphology. One comment, the foundation of the analysis relies on the percent benthic cover data presented in the raw data table. The sum of cover for all categories should equal 100%. The data table includes 585 total records, 114 of which contain no data since no sampling occurred for that location by year, and 471 records have data. Of those 471 records, 171 or 36% of the records, do not have values that sum to 100% total benthic cover. Although most of the differences are most likely due to rounding errors, this lack of attention to the most basic component of the analysis should be corrected and provides some concern about any analytical details that were not explicitly presented. I think the resistance/recovery metrics as included should be discarded. Furthermore, the comments made in prior sections should be addressed. In summary, there are interesting findings here that can be explored further. Namely, the significant declines in coral cover for many stony coral groups around 2004-2005. Examine additional coral bleaching indices to bolster your evidence that SST was a potential driver for the large declines that occurred. The loss of coral groups with complex morphologies would suggest a decline in habitat complexity from the decline in branching, tabulate, digitate corals and an increase in encrusting corals. This homogenization of coral structural complexity is a potential outcome in the decline of coral communities (Alvarez-Filip et al. (2009) Proc R Soc B 276: 3019-3025).

---

## Round 0.2 · accepted · Accept

Thank you for the thorough revisions.

#